# Study on Dynamic Changes of Soil Erosion in the North and South Mountains of Lanzhou

**Hua Zhang [1,2,*]** , **Jinping Lei [1]**, **Hao Wang [3]**, **Cungang Xu [1]** and **Yuxin Yin [1]**

1 College of Geography and Environmental Sciences, Northwest Normal University, Lanzhou 730070, China; 2020212672@nwnu.edu.cn (J.L.); cungangxu@163.com (C.X.); 2020212680@nwnu.edu.cn (Y.Y.)
2 Key Laboratory of Resource Environment and Sustainable Development of Oasis, Lanzhou 730070, China
3 Faculty of Geographic Science, Beijing Normal University, Beijing 100875, China; 18772102861@163.com
\* Correspondence: zhanghua@nwnu.edu.cn

**Abstract:** The North and South Mountains of Lanzhou City are the ecological protection barriers and an important part of the ecological system of Lanzhou City. This study takes the North and South Mountains as the study area, calculates the soil erosion modulus of the North and South Mountains of Lanzhou City based on the five major soil erosion factors in the RUSLE model, and analyses the spatial and temporal dynamics of soil erosion in the North and South Mountains of Lanzhou City and the soil erosion characteristics under different environmental factors. The results of the study show that: The intensity of soil erosion is dominated by slight erosion, which was distributed in the northwestern and southeastern parts of the North and South Mountains in 1995, 2000, 2005, 2010, 2015 and 2018. Under different environmental factors, the soil erosion modulus increased with elevation and then decreased; the soil erosion modulus increased with a slope; the average soil erosion modulus of grassland was the largest, followed by forest land, cultivated land, unused land, construction land, and it was the smallest for water; except for bare land, the average soil erosion modulus decreases with the increase of vegetation cover; Soil erosion modulus was the greatest in the pedocal of the North and South Mountains, and the least in the alpine soil.

**Keywords:** soil erosion; RUSLE model; erosion intensity; land desertification; North and South Mountains of Lanzhou

## 1. Introduction

Soil erosion is the destruction and loss of soil and water resources and land productivity due to natural forces and human activities, mainly including land surface erosion and water loss, which is the most dominant form of soil degradation [1–3]. Soil erosion will destroy the surface structure, reduce land fertility, raise the riverbed, destroy water conservancy facilities, aggravate flood and drought, and pose a significant threat to agricultural production, river water quality, and the environment. Soil erosion has become one of the world's most extensive and complicated ecological problems. It has become the concern of many disciplines [4], such as soil science, agronomy, hydrology, environmental science, and so on [5–8]. China is one of the countries with the most severe soil erosion [9,10]. In 2018, the soil erosion area in China reached $2.73 \times 10^6$ km², accounting for about 28.80% of the total area in China except Taiwan Province, a large area and a wide distribution [11]. The area of soil erosion in Northwest China is $1.26 \times 10^6$ km², accounting for 40.95% of the total area of Northwest China, and this presents a fundamental environmental problem [12]. In Gansu Province, for instance, the soil erosion area reached $1.86 \times 10^5$ km², accounting for about 40.66% of the total area, which exerted significant pressure on soil and water conservation and the construction of ecological civilization.

Lanzhou City, the capital of Gansu Province, is located in the upper basin of the Yellow River and consists of a pearl-shaped basin formed by the alluvial deposits of the Yellow River. To the north and south of Lanzhou City are the North and South Mountains, a

mountain range covered with loess formed by the terraces of the Yellow River. To the north of Lanzhou City is the Tengger Desert; to the west is the Badanjilin Desert, a region with $2.78 \times 10$ km$^2$ of severely desertified land [13], resulting in frequent sandstorms, serious surface exposure and fragile ecosystems [14–16]. The natural vegetation in the North and South Mountains of Lanzhou is mainly desert vegetation, at present, the area of the environmental greening project in the North and South Mountains of Lanzhou City has reached 413 km$^2$, with $1.6 \times 10^8$ trees of various types being established, forming a complete artificial ecosystem and making the North and South Mountains an important ecological barrier [17,18]. Therefore, it is of great significance for soil and water conservation and ecological civilization construction in Lanzhou to reveal the temporal and spatial characteristics of soil erosion and analyze the dynamic changes of soil erosion in Lanzhou.

Soil erosion models are a common method for quantitative soil erosion estimation. USDA and university cooperations saw the establishment of the Revised Universal Soil Loss Equation (RUSLE), based on the Universal Soil Loss Equation (USLE) in 1986 [19], which has become a widely used model for quantitative soil erosion estimation worldwide due to its simplicity, few parameter requirements, and high estimation accuracy compared to other soil erosion models [20–23]. Therefore, this study takes the South and North Mountains of Lanzhou City as the research area, takes soil erosion as the research content, and uses the RULSE model to calculate the soil erosion modulus of the South and North Mountains of Lanzhou City based on soil field sampling data, land use, and precipitation data. The objectives of this study are (1) to reveal the spatial and temporal variation characteristics of soil erosion in the South and North Mountains of Lanzhou City, and (2) to provide scientific reference for the construction of water and soil conservation and ecological civilization.

## 2. Materials and Methods

### 2.1. General Situations of the Study Area

The North and South Mountains of Lanzhou span Anning District, Qilihe District, Chengguan District, Xigu District, Gaolan County, and Yuzhong County within the jurisdiction of Lanzhou City, with geographical coordinates of 35°44'–36°19' N, 103°21'–103°59' E. The total area is about 1940.08 km$^2$ (Figure 1). Among them, the green part accounts for 846.66 km$^2$, and the non-green region accounts for 1147.42 km$^2$. The geological conditions of this area are involved. The topography is fragmented, and natural disasters are to occur easily. The climate type belongs to the temperate semi-arid continental monsoon climate, with an annual average temperature of 9.1 °C and a yearly average rainfall of 327.7 mm, mostly concentrated from July to September, and the average annual potential evaporation is 1468 mm, which is 4.4 times of the precipitation. The vegetation type belongs to the transition type of typical steppe to desert steppe. Currently, most of the existing forests in Lanzhou's North and South Mountains are artificial forests, mostly young and middle-aged. Artificial afforestation is mainly coniferous and broad-leaved mixed forest, arbor shrub mixed forest, and shrub forest. The soil types in this area are primarily grey calcareous soil, mostly dark grey calcareous soil and typically grey calcareous soil in the South Mountains, and light grey calcareous soil and red sandy soil in the northern mountain, with a loose texture and weak anti-erosion ability.

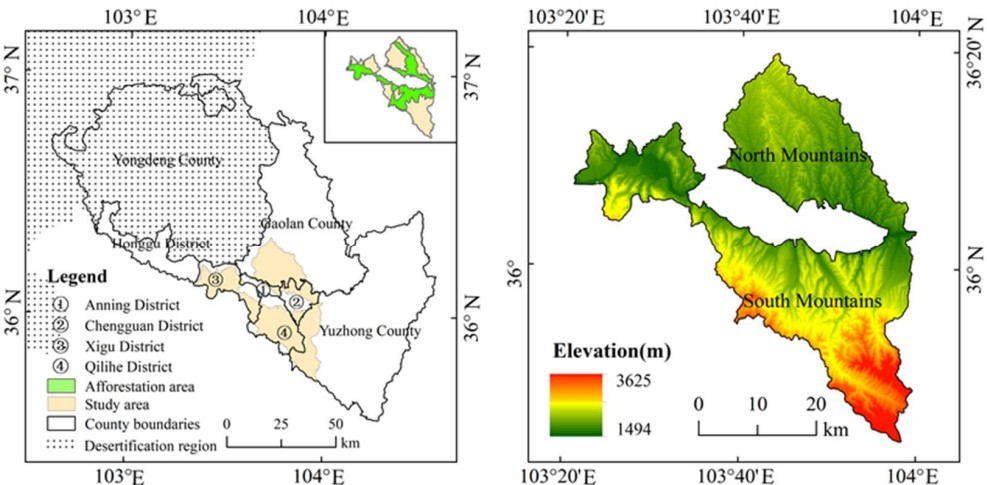

**Figure 1.** Overview of the study area.

*2.2. Data Source*

2.2.1. Soil Texture and Organic Carbon Data

(1)    Soil sample sampling

Using the 1:1 million soil map of the North and South Mountains of Lanzhou City as the base map, about 120 soil sampling points were designed in July–August 2019 according to a uniform distribution method of 4 km × 4 km, The sampling was carried out according to the plan, combining the actual situation with randomly selected 10 m × 10 m sample plots. A total of 130 soil samples were collected. The field sampling operation was carried out using a soil auger to collect soil samples from 0–20 cm of the surface layer at the center of the sample plots and four right-angle points, mixed evenly and placed in self-sealing bags for the determination of soil texture and soil organic carbon. 0–20 cm soil samples of the surface layer at the center and four right corners were collected with a ring knife, put into an aluminum box, and weighed fresh at the sampling site, which was used to determine the soil bulk density. Using GPS positioning, the elevation, longitude, and latitude of the sampling points in the center were recorded and numbered sequentially.

(2)    Determination of soil samples

The determination of soil texture was carried out by the Mastersizer 2000 laser particle size analyzer (model: MS2000, Zhenxiang Technology Co., Ltd., Changsha, China). The soil organic carbon content was determined by the Qiulin method, the soil salinity was determined by the "residue drying-mass method", and the pH value was determined by the "potential method".

2.2.2. Other Data

(1) The meteorological data was based on the monthly precipitation data set of 0.5° × 0.5° in China from 1995 to 2018 (V2.0), which came from the China Meteorological data sharing Network (http://data.cma.cn/), (accessed on 25 March 2019). (2) The GDEM-DEM 30 m spatial resolution digital elevation was derived from the geospatial data cloud (http://www.gscloud.cn/), (accessed on 25 March 2019). (3) The Landsat TM/OLI image from 1995 to 2018 was selected as the source of the Google Earth Engine cloud platform, to calculate the Normalized Difference Vegetation Index (NDVI) for the study area (Google Earth Engine, GEE) (https://earthengine.google.com/), (accessed on 25 March 2019). The image was programmed in the platform to preprocess the image. (4) The land use with a spatial resolution of 30 m in 1990, 2000, 2005, 2010, 2015, and 2018 was selected from the Resource and Environmental Science Data Center of the Chinese Academy of Sciences (http://www.resdc.cn/), (accessed on 25 March 2019). (5) National Cryosphere Desert Data

Center provided the desertification distribution data (http://www.ncdc.ac.cn), (accessed on 25 March 2019).

*2.3. Research Methods*

2.3.1. Soil Erosion Model

The study used the RUSLE model to estimate soil erosion in Lanzhou [24–26]. The formula is as follows:

$$A = R \cdot K \cdot LS \cdot C \cdot P \tag{1}$$

Among them, $A$ is the average soil erosion amount per unit area last year, the unit is $[t/(km^2 \cdot a)]$, and $R$ is the precipitation erosivity factor, the unit is $[MJ\ mm/(km^2 \cdot h \cdot a)]$, $K$ is the soil erodibility factor, in units $[t \cdot km^2 \cdot h/(km^2 \cdot MJ \cdot mm)]$, $LS$ is the slope length factor (dimensionless); $C$ is the vegetation cover and management factor, (dimensionless); $P$ is the soil and water conservation and measure factor (dimensionless).

2.3.2. Determination of Factors in the RUSLE Model

(1)　Determination of $R$-value of precipitation erosivity factor

Precipitation is one of the important exogenous forces causing soil erosion, reflecting the potential impact of annual average or maximum precipitation on soil erosion. This study adopted the method of estimating rainfall erosivity by using yearly and monthly precipitation data proposed by Wischmeier [27]. The formula is as follows:

$$R = \sum_{i=1}^{12} \left[ 1.735 \times 10^{\left( 1.5 \times \log \frac{P_i^2}{P} - 0.8188 \right)} \right] \tag{2}$$

In the formula, $P_i$ is monthly precipitation (mm); $P$ is annual precipitation (mm). This method has been applied in the western region, and good results have been obtained km [28].

(2)　K value of soil erodibility factor

The soil erodibility factor refers to the soil loss rate under a given unit of precipitation erosivity measured in a standard plot [29,30]. In this study, Williams' calculation method of soil erodibility factor $K$ in the EPIC model was adopted [31]. The formula is as follows:

$$
\begin{aligned}
K = & \ 0.1317 \times \left\{ 0.2 + 0.3 \exp \left[ -0.0256\ sand \left( 1 - \frac{slit}{100} \right) \right] \right\} \times \left[ \frac{silt}{clay + silt} \right]^{0.3} \\
& \times \left[ 1 - \frac{0.25C}{C + \exp(3.72 - 2.95C)} \right] \times \left[ 1 - \frac{0.7sn1}{sn1 + \exp(-5.51 + 22.9\ sn1)} \right]
\end{aligned} \tag{3}
$$

Among them, *Sand*, *Silt*, and *Clay* represent the percentage of sand, silt, and clay content in soil, respectively (%); $C$ is the percentage of soil organic carbon content (%); $Sn1 = 1 - Sand/100$. Generally, a higher value of the soil erodibility factor $K$ indicates that the soil is poorly resistant to erosion and susceptible to erosion; conversely, the soil is not susceptible to erosion [32–34].

(3)　km LS value of slope length factor

The slope length factor, i.e., the topographic factor, determines the state and direction of movement of surface runoff [35]. The greater the slope and the longer the slope length, the greater the potential energy that surface runoff will acquire, and the more intense the erosive effect on soil. In this study, the slope and slope length factors were extracted by the formulas studied by McCool et al. [36] and Liu et al. [37]. The calculation formulae of slope factors are as follows:

$$S = \begin{cases} 10.8 \cdot \sin\theta + 0.03 & \theta < 6 \\ 16.8 \cdot \sin\theta - 0.50 & 5 \le \theta < 14 \\ 21.91 \cdot \sin\theta - 0.90 & \theta < 14 \end{cases} \tag{4}$$

where S is the slope factor (dimensionless), and $\theta$ is the slope value (°), which can be extracted from the DEM data.

The formula for calculating the slope length factor is as follows:

$$L = (\lambda/22.13)^\alpha \tag{5}$$

$$\lambda = flowacc \times cellsize \tag{6}$$

$$\alpha = \beta/(1+\beta) \tag{7}$$

$$\beta = (\sin\theta/0.089)/[3.0 \times (\sin\theta)^{0.8} + 0.56] \tag{8}$$

Among them, $L$ is the slope length factor, and its value is the amount of soil erosion produced on the standard slope of 22.13 m. The $\lambda$ is the slope length, where *flowacc* is the catchment accumulation, *cellsize* is the size of the DEM data grid pixel, and $\alpha$ is the slope length, $\theta$ is the slope value, in units of (°); $\beta$ is the parameter that determines $\alpha$.

(4)  C value of vegetation cover and management factor

Vegetation can protect the surface soil and slow down the rate of soil erosion [38]. NDVI is the most common data to calculate the C value of vegetation cover and management factor [39]. The NDVI number 9 used in this study is derived from the Google Earth Engine cloud platform, and the formula proposed by VanderKnijff et al. [40] is used to calculate the C value of vegetation cover and management factor. The formula is as follows:

$$C = \exp\left[-a \times \frac{NDVI}{b - NDVI}\right] \tag{9}$$

Among them, $C$ is the vegetation cover and management factor (dimensionless); $a$ and $b$ are the parameters that determine the NDVI-C relationship curve. Through VanderKnijff experiments, it is found that the most appropriate values are $a = 2$ and $b = 1$. This method has been studied in China and has achieved good results. According to the Formula (9), if the C value is negative, the assignment is 0 for all negative values; if the C value is greater than 1, the assignment is 1 for all values greater than 1. The higher the C value, the worse the vegetation growth; on the contrast, the lower the C value, the better the vegetation growth.

(5)  *p*-value of soil and water conservation measures

The factor of soil and water conservation and measures generally refers to the ratio of the amount of soil loss when certain engineering measures are taken in a certain area to the amount of soil loss without engineering measures under the same conditions. Its value ranges from 0 to 1; 0 means that soil erosion will not occur in this area, and 1 means no soil and water conservation measures have been taken [38,41–43].

## 3. Results

### 3.1. Calculation of Each Factor in the RUSLE Model

(1)  R-value of precipitation erosivity factor

This method has been applied in the western region, and good results have been obtained [28]. The average precipitation erosivity factors in 1995, 2000, 2005, 2010, 2015, and 2018 in Lanzhou were 110.06, 83.20, 71.09, 46.68, 56.97 and 198.61 [MJ·mm/(km²·h·a)] respectively. Spatially, the precipitation erosivity factors of the North and South Mountains decreased from southeast to northwest in 1995, 2000, 2005, and 2010. The precipitation erosivity factors of the North and South Mountains decreased from the west to the east in 2015 and 2018. The precipitation erosivity factor of the west was greater than that of the east. The erosivity factor of precipitation in 2018 was significantly higher than in other years, mainly because 2018 was an abnormally rainy year. The precipitation was higher than that in previous years (Figure 2).

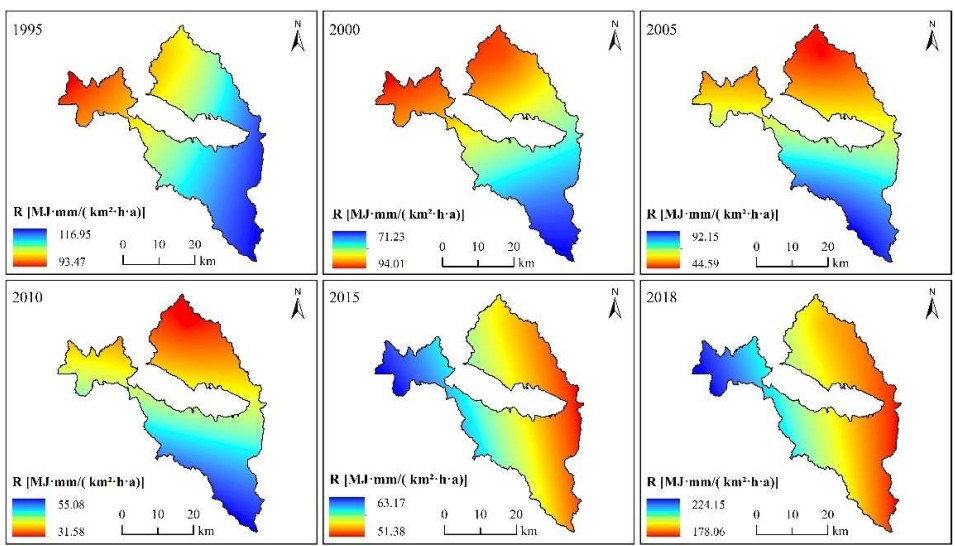

**Figure 2.** Spatial distribution of rainfall erosivity in the South and North Mountains of Lanzhou in 1995, 2000, 2005, 2010, 2015 and 2018.

(2)    K value of soil erodibility factor

According to the data of soil texture and soil organic carbon content of the sampling points, the K value was calculated, and ordinary kriging interpolation was performed in ArcGIS 10.4 software. The spatial distribution of soil erodibility factors in the northern and southern mountains of Lanzhou City was calculated according to Formula (3) (Figure 3). The areas with a soil erodibility factor of 0.054–0.061 $t\cdot km^2\cdot h/(km^2\cdot MJ\cdot mm)$ were mainly distributed in the central and eastern regions, and the areas with ? soil erodibility factor of 0.045–0.053 $t\cdot km^2\cdot h/(km^2\cdot MJ\cdot mm)$ were mainly distributed in the western, northwest and southern regions. The areas with a soil erodibility factor of 0.037–0.044 $t\cdot km^2\cdot h/(km^2\cdot MJ\cdot mm)$ were mainly distributed in parts of the North Mountain. The areas with a soil erodibility factor of 0.018–0.036 $t\cdot km^2\cdot h/(km^2\cdot MJ\cdot mm)$ were mainly distributed in the western part of the North Mountain.

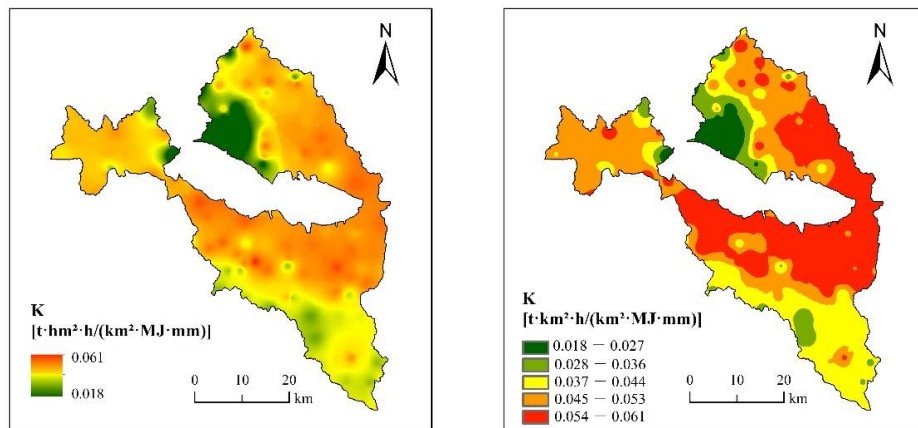

**Figure 3.** Spatial distribution of soil erodibility factor K in the South and North Mountains of Lanzhou.

(3)    LS value of slope length factor

The spatial distribution of the slope factor of the North and South Mountains in Lanzhou (Figure 4) showed that the minimum value of the slope factor was 0, the maximum value was 58.98, the average value was 15.52, the minimum value of the slope factor was 0, the maximum value was 9.99, the average value was 4.76, the minimum value of the

slope length factor was 0, the maximum value was 5.92, and the average value was 2.22. The minimum value of the slope length factor was 0, the maximum value was 59.19, and the average value was 12.20. The overall upper slope, slope factor, slope length factor, and slope length factor were zonal distributions, and the slope length factor of the South Mountain was larger than that of the North Mountain.

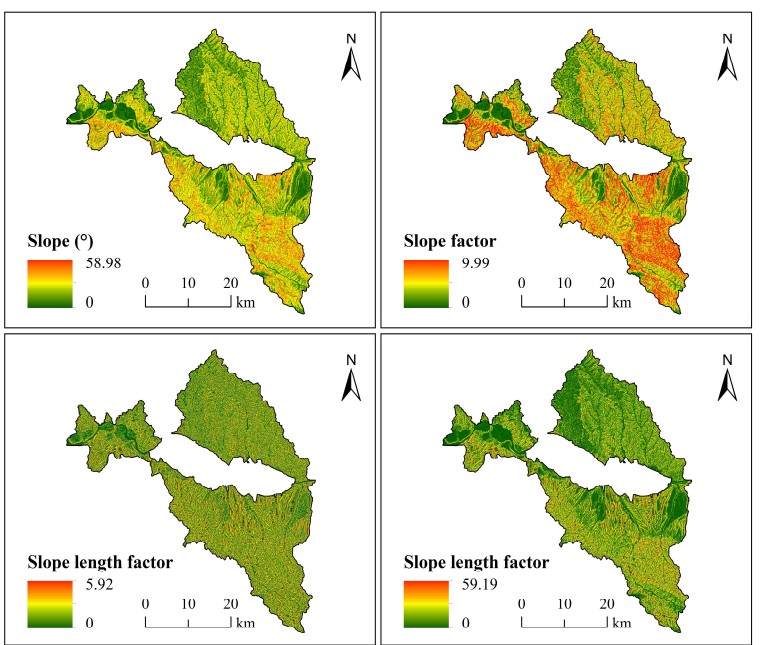

**Figure 4.** Spatial distribution of the gradient slope and slope length factor in the Northand South Mountains of Lanzhou.

(4)    C value of vegetation cover and management factor

The average values of vegetation cover and management factors in the North and South Mountains of Lanzhou in 1995, 2000, 2005, 2010, 2015, and 2018 were 0.34, 0.43, 0.56, 0.50, 0.40 and 0.57, respectively. Overall, the vegetation cover and management factors were the lowest in 1995 and the highest in 2018. The C value of the North Mountain was higher than that of the South Mountain, indicating that the vegetation coverage of the North Mountain was lower than that of the South Mountain (Figure 5).

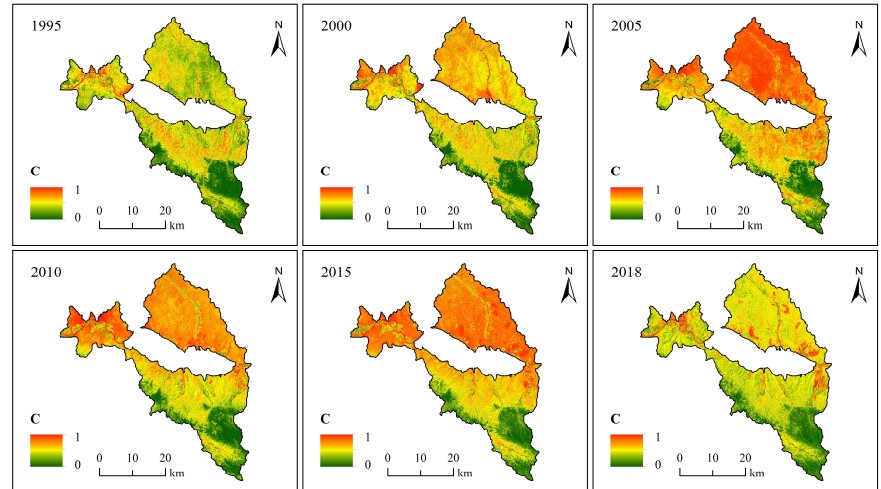

**Figure 5.** Spatial distribution of vegetation cover and management factors in the South and North Mountains of Lanzhou in 1995, 2000, 2005, 2010, 2015 and 2018.

(5)    *p*-value of soil and water conservation measures

In this study, according to Table 1, the land use data of 1995, 2000, 2005, 2010, 2015, and 2018 were assigned, and the spatial distribution of the *p*-value of soil and water conservation measure factors with 30 m resolution was obtained (Figure 6). As the land-use change in Lanzhou's North and South Mountains was not obvious, the spatial distribution of soil and water conservation measures in the North and South Mountains was consistent, and the change was not obvious.

**Table 1.** *p* values of different land-use types in the South and North Mountains of Lanzhou.

| Land-Use Type | Cultivated Land | Forest Land | Grassland | Water Area | Construction Land | Unused Land |
| --- | --- | --- | --- | --- | --- | --- |
| *p* | 0.35 | 1.0 | 1.0 | 0.0 | 0.0 | 1.0 |

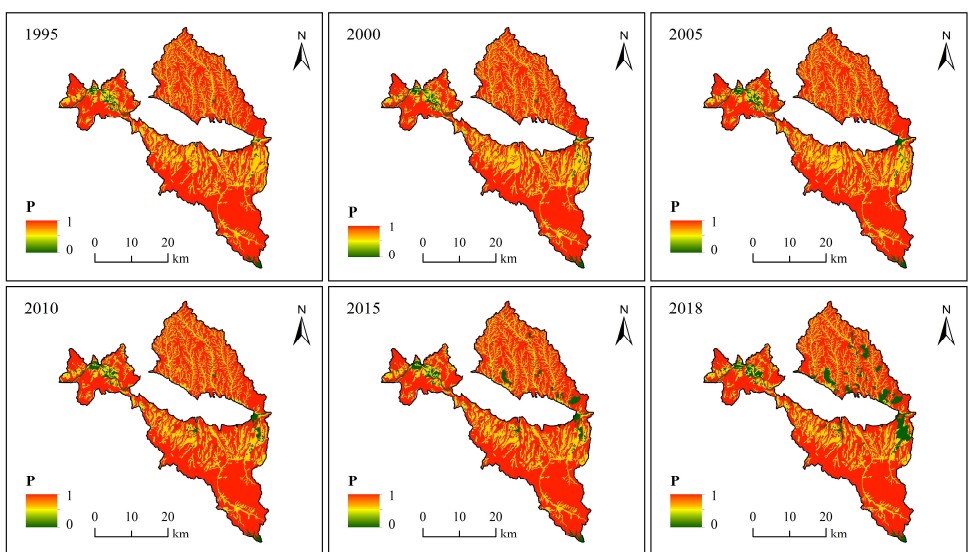

**Figure 6.** Spatial distribution of soil and water conservation measures factors in the South and North Mountains of Lanzhou in 1995, 2000, 2005, 2010, 2015 and 2018.

### 3.2. Spatio-Temporal Variation Characteristics of Soil Erosion

According to the soil erosion modulus in different years, the average soil erosion modulus in the North and South Mountains of Lanzhou City generally showed a first fluctuating downward trend and started to increase after 2016, and in 2018 there was an abrupt change and a sudden increase to 25.83 t/(km$^2$·a). The annual average soil erosion amount was $330.74 \times 10^4$ t (1995), $323.80 \times 10^4$ t (2000), $342.09 \times 10^4$ t (2005), $200.20 \times 10^4$ t (2010), $314.41 \times 10^4$ t (2015), and $515.14 \times 10^4$ t (2018) (Figure 7).

According to the Ministry of Water Resources (SL190-2007) Soil Erosion Classification and Grading Standard [44], the study area is divided into six soil erosion intensity classes according to the soil erosion modulus, namely, slight erosion [5–25 t/(km$^2$·a)], light erosion [5–25 t/(km$^2$·a)], moderate erosion [25–50 t/(km$^2$·a)], strong erosion [50–80 t/(km$^2$·a)], extremely strong erosion [80–150 t/(km$^2$·a)] and severe erosion [150 t/(km$^2$·a)]. The spatial distribution of soil erosion intensity classes for 1995, 2000, 2005, 2010, 2015 and 2018 was obtained for the North and South Mountains of Lanzhou (Figure 8). Soil erosion intensity in the North and South Mountains of Lanzhou in 1995, 2000, 2005, 2010, 2015, and 2018 was mainly slight erosion. The amount of soil erosion in 2018 was significantly higher than that in other years, mainly because 2018 was an unusually wet year, and the precipitation was higher than in previous years. It was mainly distributed in the northwest and southeast of the North and South Mountains. The strong, extremely strong, and severe soil erosion was mainly distributed in the middle of the South Mountain and the middle of the North Mountain (Figure 8).

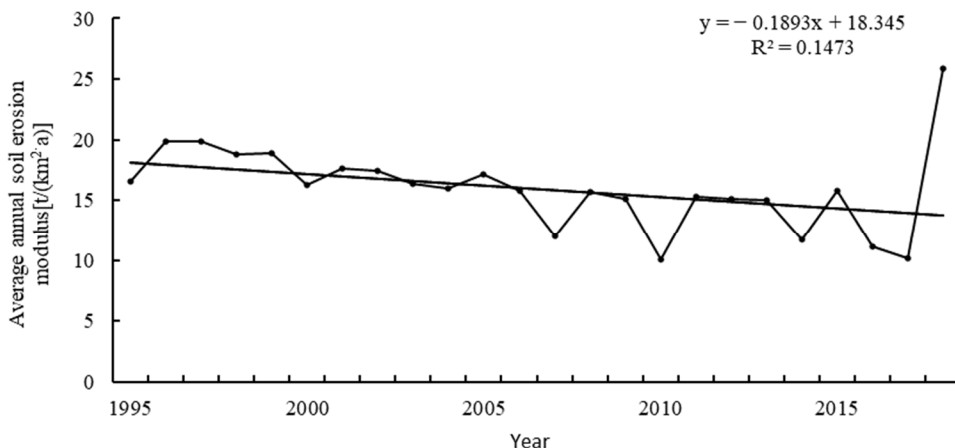

**Figure 7.** Time change of annual soil erosion modulus in the Northand SouthMountains of Lanzhou from 1995 to 2018.

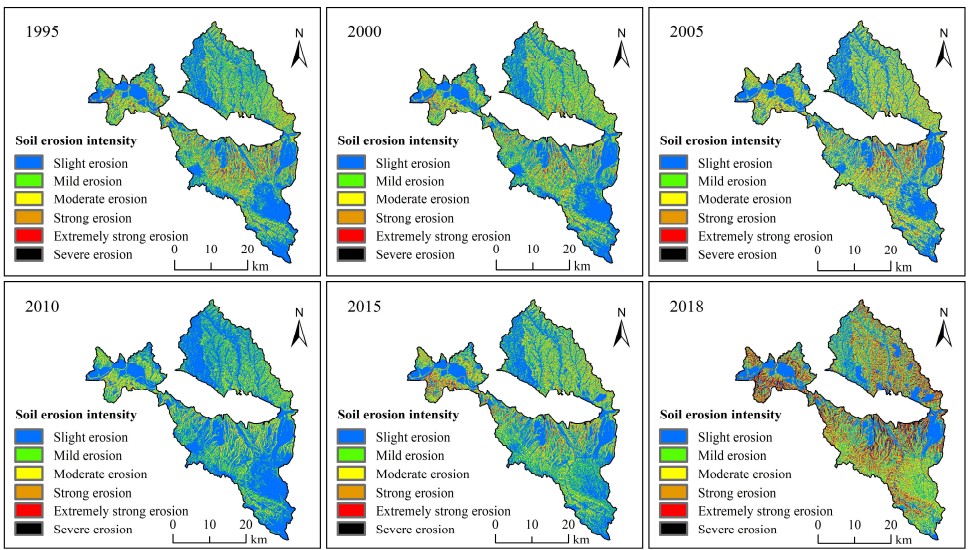

**Figure 8.** Spatial distribution of soil erosion intensity in the South and North Mountains of Lanzhou in 1995, 2000, 2005, 2010, 2015 and 2018.

### 3.3. Area Transfer Characteristics of Soil Erosion Intensity

Based on the statistical analysis of the data of different erosion intensity areas, the transfer chord diagram of soil erosion intensity was obtained. From 1995 to 2018, the area of soil slight erosion was the largest, which was 735.92 km$^2$, followed by mild, moderate, strong, extremely strong, and the area of severe erosion remained the smallest. Among them, slight soil erosion mainly shifted to mild soil erosion. The mild soil erosion transferred to slight soil erosion, and the area was 215.14 km$^2$; moderate soil erosion mainly shifted to slight and mild soil erosion; strong soil erosion mainly shifted to mild and moderate soil erosion; moderate and strong soil erosion transferred to extremely strong soil erosion; severe soil erosion shifted to strong and extremely strong soil erosion. From 1995 to 2018, the stability rates of slight, mild, moderate, strong, extremely strong and severe soil erosion were 36.91%, 4.64%, 2.74%, 0.79%, 0.77% and 0.08%, respectively (Figure 9).

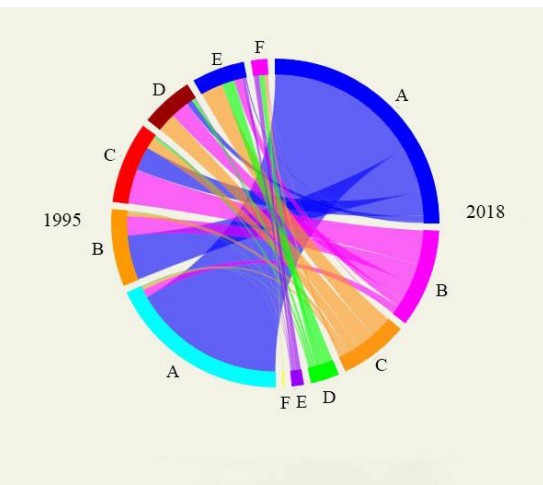

**Figure 9.** Chordal graph of soil erosion intensity in South and North Mountains of Lanzhou from 1995 to 2018 (Note: A: slight erosion; B: mild erosion; C: moderate erosion; D: strong erosion; E: extremely strong erosion; F: severe erosion).

*3.4. Characteristics of Soil Erosion under Different Environmental Factors*

3.4.1. Characteristics of Soil Erosion at Different Elevations

The soil erosion modulus in 1995, 2000, 2005, 2010, 2015, and 2018 was superimposed and analyzed according to different elevations, and the average soil erosion modulus at different elevations was obtained (Table 2). It can be seen from the table that the soil erosion modulus first increased and then decreased with the increase in altitude; at the height of 1494–1800 m, the slope length factor is lower, so the soil erosion modulus is lower. At the height of 1800–2100 m, there are more gullies, the slope length factor is high, andcoupled with human interference, so the soil erosion modulus is the largest. With the increase in altitude, the growth of vegetation is better, the vegetation cover and management factors are lower, and coupled with the reduction of human activities, the degree of soil intervention is low. Accordingly, the average soil erosion modulus decreased with altitude.

**Table 2.** Modulus of soil erosion in different years at different altitudes in the South and North Mountains of Lanzhou (unit: t/(km$^2$·a)).

| Elevation (m) | 1995 | 2000 | 2005 | 2010 | 2015 | 2018 | Average Value |
|---|---|---|---|---|---|---|---|
| 1494–1800 | 15.05 | 15.40 | 15.35 | 9.92 | 15.11 | 31.86 | 17.11 |
| 1800–2100 | 20.82 | 19.73 | 20.44 | 12.00 | 19.03 | 41.21 | 22.21 |
| 2100–2400 | 18.21 | 16.58 | 17.95 | 9.48 | 15.07 | 40.28 | 19.59 |
| 2400–2700 | 10.36 | 11.25 | 14.51 | 6.98 | 11.13 | 32.45 | 14.45 |
| 2700–3000 | 6.81 | 7.66 | 11.03 | 4.88 | 8.41 | 24.02 | 10.47 |
| 3000–3300 | 2.92 | 3.95 | 8.20 | 3.07 | 6.52 | 19.35 | 7.33 |
| 3300–3625 | 1.26 | 1.83 | 3.46 | 1.56 | 2.60 | 6.73 | 2.91 |

3.4.2. Characteristics of Soil Erosion under Different Slopes

The soil erosion modulus in 1995, 2000, 2005, 2010, 2015, and 2018 was analyzed according to different slope grades, and the average soil erosion modulus under different slopes was obtained (Table 3). On the whole, the average soil erosion modulus was the highest on the slope of >35°and the lowest on the slope of 0–5°. The 17 modulus of soil erosion increased with the increase in slope. This is mainly because the higher the slope, the greater the slope factor, and the rapid erosion of the runoff velocity caused by surface water is serious.

**Table 3.** Modulus of soil erosion in different years under different slopes in the South and North Mountains of Lanzhou (unit: t/(km$^2$·a)).

| Slope | 1995 | 2000 | 2005 | 2010 | 2015 | 2018 | Average Value |
|---|---|---|---|---|---|---|---|
| 0–5° | 4.19 | 3.85 | 3.82 | 2.20 | 3.47 | 8.08 | 4.27 |
| 5–8° | 7.09 | 6.65 | 6.62 | 3.82 | 5.98 | 13.74 | 7.32 |
| 8–15° | 13.55 | 13.07 | 13.28 | 7.66 | 11.94 | 27.10 | 14.43 |
| 15–25° | 21.40 | 21.13 | 22.32 | 13.08 | 20.45 | 46.46 | 24.14 |
| 25–35° | 27.44 | 27.31 | 30.17 | 17.95 | 28.58 | 65.04 | 32.75 |
| 35–60° | 31.46 | 31.07 | 35.47 | 20.90 | 33.40 | 76.90 | 38.20 |

### 3.4.3. Characteristics of Soil Erosion under Different Land-Use Types

The average soil erosion modulus under different land-use types was obtained based on the regional statistical analysis of land use classification and soil erosion modulus in 1995, 2000, 2005, 2010, 2015, and 2018 (Table 4). Overall, the average soil erosion modulus of grassland and woodland was larger, which is 24.76 t/(km$^2$·a) and 23.43 t/(km$^2$·a), respectively. The average soil erosion modulus of cultivated land was 7.48 t/(km$^2$·a), and the average soil erosion modulus of water area was 0.27 t/(km$^2$·a), which is the smallest. Although grassland and woodland are covered by vegetation, grassland and woodland are relatively high above sea level, generally distributed in areas with high mountains and steep slopes, and soil erosion is more serious due to water runoff and gravity.

**Table 4.** Modulus of soil erosion in different years under different land-use types in the South and North Mountains of Lanzhou (unit: t/(km$^2$·a)).

| Land-Use Type | 1995 | 2000 | 2005 | 2010 | 2015 | 2018 | Average Value |
|---|---|---|---|---|---|---|---|
| Cultivated land | 7.51 | 6.87 | 6.83 | 3.71 | 5.71 | 14.26 | 7.48 |
| Grassland | 21.12 | 22.37 | 22.42 | 13.30 | 21.09 | 48.25 | 24.76 |
| Woodland | 19.31 | 20.87 | 20.97 | 11.52 | 19.38 | 48.53 | 23.43 |
| Water area | 0.32 | 0.24 | 0.13 | 0.11 | 0.23 | 0.57 | 0.27 |
| Construction land | 0.47 | 0.37 | 0.32 | 0.19 | 0.22 | 0.60 | 0.36 |
| Unused land | 1.18 | 1.20 | 1.32 | 0.58 | 0.48 | 2.68 | 1.24 |

### 3.4.4. Characteristics of Soil Erosion under Different Vegetation Coverage

Based on the regional statistical analysis of soil erosion modulus in 1995, 2000, 2005, 2010, 2015, and 2018 according to the different vegetation coverage, the average soil erosion modulus under different land-use types was obtained (Table 5). On the whole, the soil erosion modulus was the largest under low mulch. Except for bare land, the average soil erosion modulus decreased with the increase in vegetation coverage. The rise of vegetation coverage reduced rain water's splashing and running water scouring. The roots of vegetation maintained the soil and played a role in slowing down soil erosion. The average soil erosion modulus of bare land ranked third, mainly because according to the vegetation coverage of less than 10%, and the land use types are construction land and unused land., construction land was generally cement-hardened and difficult to erode; unused land generally had poor soil texture and relatively weak erosion.

**Table 5.** Modulus of soil erosion in different years under different vegetation coverage in the South and North Mountains of Lanzhou (unit: t/(km²·a)).

| Vegetation Coverage | 1995 | 2000 | 2005 | 2010 | 2015 | 2018 | Average Value |
|---|---|---|---|---|---|---|---|
| Bare land | 21.67 | 17.29 | 16.52 | 11.31 | 19.66 | 21.05 | 15.50 |
| Low vegetation cover | 21.21 | 19.70 | 21.90 | 12.92 | 18.78 | 43.90 | 20.06 |
| Medium and low vegetation cover | 19.03 | 19.71 | 20.82 | 11.12 | 15.48 | 44.21 | 19.05 |
| Medium vegetation cover | 14.42 | 12.68 | 16.60 | 7.58 | 11.19 | 38.34 | 14.97 |
| Medium and high vegetation cover | 9.19 | 7.88 | 12.35 | 5.51 | 9.55 | 31.47 | 11.56 |
| High vegetation cover | 3.36 | 3.16 | 5.16 | 2.73 | 7.18 | 21.64 | 7.03 |

### 3.4.5. Soil Erosion Characteristics under Different Soil Types

In ArcGIS 10.4, the South and North Mountains of Lanzhou City were classified into semi-luvisols, primarosol, pedocal, xerosol, alpine soil and anthrosol according to the definition criteria of the Chinese soil outline [45]. The soil erosion moduli in 1995, 2000, 2005, 2010, 2015 and 2018 were superimposed and analysed according to different soil types to obtain the average soil erosion moduli under different soil types. The average soil erosion modulus was highest for pedocal and lowest for alpine soil (Table 6).

**Table 6.** Modulus of soil erosion in different years at different soil type in the South and North Mountains of Lanzhou City (unit: t/(km²·a)).

| Soil Type | 1995 | 2000 | 2005 | 2010 | 2015 | 2018 | Average |
|---|---|---|---|---|---|---|---|
| semi-luvisols | 8.47 | 9.33 | 12.77 | 5.79 | 9.89 | 29.00 | 12.54 |
| primarosol | 17.94 | 17.93 | 18.43 | 11.38 | 17.50 | 38.31 | 20.25 |
| pedocal | 21.63 | 17.60 | 21.50 | 10.75 | 15.86 | 41.18 | 21.42 |
| xerosol | 14.10 | 15.68 | 14.60 | 8.76 | 15.31 | 31.77 | 16.70 |
| alpine soil | 4.40 | 6.43 | 10.12 | 4.59 | 6.58 | 17.83 | 8.32 |
| anthrosol | 15.21 | 12.35 | 11.18 | 8.46 | 12.67 | 27.56 | 14.57 |

### 3.4.6. Comparison of Soil Erosion Inside and Outside the Environmental Greening Project

According to the statistics of soil erosion modulus in 1995, 2000, 2005, 2010, 2015 and 2018, the average soil erosion modulus were inside and outside the environmental greening project (Table 7). The average soil erosion modulus inside and outside the environmental greening project was 21.27 t/(km² a) and 23.56 t/(km² a), respectively, and the average soil erosion modulus outside the environmental greening project was larger than that inside the environmental greening project. The area of slight soil erosion inside and outside the greening area was the largest, and the area of severe soil erosion was the smallest, which was 7.72 km² and 4.11 km², respectively, accounting for 0.39% and 0.21% of the total area. Within and outside the greening range, the area occupied by soil erosion intensity from large to small was slight, mild, moderate, strong, extremely strong and severe, respectively (Figure 10).

**Table 7.** Modulus of soil erosion in different years inside and outside the environmental greening project in the South and North Mountains of Lanzhou City (unit: t/(km²·a)).

| | 1995 | 2000 | 2005 | 2010 | 2015 | 2018 | |
|---|---|---|---|---|---|---|---|
| inside | 18.81 | 18.55 | 19.74 | 11.98 | 18.36 | 40.19 | 21.27 |
| outside | 22.28 | 21.79 | 22.19 | 15.29 | 20.57 | 39.23 | 23.56 |

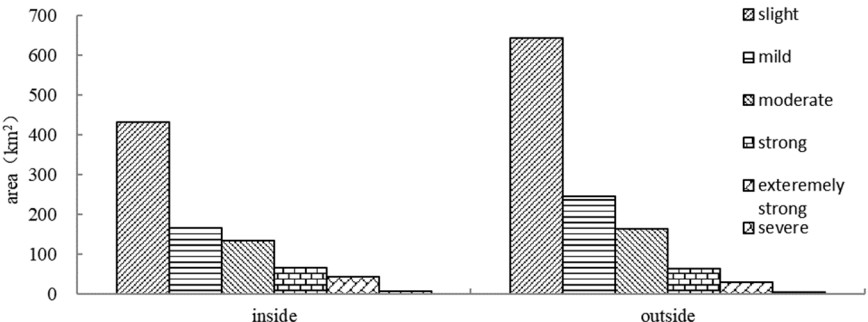

**Figure 10.** Average area of soil erosion intensity inside and outside the environmental greening project of South and North Mountains in Lanzhou City (unit: km$^2$).

## 4. Discussion

From 1995 to 2018, the average soil erosion modulus in the North and South Mountains of Lanzhou showed a fluctuating downward trend. There was an abnormality in 2018, mainly due to precipitation. The natural precipitation in the North and South Mountains of Lanzhou was less and the seasonal distribution is uneven. The precipitation was mostly concentrated in July, August, and September, and its rainfall accounts for 60–70% of the annual rainfall. However, 2018 was a year of abnormal increase in precipitation. Although the average precipitation was 441.6 mm, which is less than that in other areas, the rainfall erosivity was the highest due to the low vegetation coverage and the collapsibility of loess. Torrential rain caused flash floods, landslides, and mudslides, resulting in the largest in soil erosion modulus in 2018.

Due to climate, natural disasters, and man-made deforestation, soil erosion and desertification in the study area are becoming more and more serious. Lanzhou's urban construction is closely related to the ecological construction of the North and South Mountains. In order to improve its ecological environment, Lanzhou started planting plantation greening projects in 1999. Therefore, most of the existing forests in Lanzhou are plantations, and a preliminary plantation ecosystem has been formed. At present, the method of increasing vegetation coverage and reducing the bare area of the ground has achieved initial results, and the soil erosion modulus within the greening project is lower than that outside the greening project.

Soil erosion moduli can be determined in a variety of ways, such as using measured runoff sediment information, simulated rainfall, field surveys, radioisotopes, and mathematical models. In this paper, the soil erosion modulus was calculated using the Revised Universal Soil Loss Equation (RUSLE). Although the limitations of the application of RUSLE in the North and South Mountions of Lanzhou City were corrected to the greatest extent possible, the RUSLE can only calculate the hydraulic erosion modulus, and parts of the North and South Mountains are in a wind erosion zone with a large area of desertification and dust storms occurring every spring [46]. As there is still a lack of effective calculation models for wind and water erosion composite soil erosion modulus, the changes in wind erosion modulus are not considered in this paper, resulting in a small calculation of soil erosion modulus in windy and sandy areas. Soil erosion is closely related to land desertification, and effective management of soil erosion can help to slow down the development of land desertification in the study area [47,48].

Countermeasures to prevent soil erosion in the North and South Mountains are the following:

(1) Reasonably control the development of land resources in the North and South Mountains and improve the compensation mechanism for land resource protection.

(2) Strengthen the construction of scientific field observation and research stations for the ecological environment in the North and South Mountains of Lanzhou.

(3)   Increase investment in environmental greening project in the North and South Mountains and consolidate the role of the ecological security barrier in the North and South Mountains.

## 5. Conclusions

(1)   The average precipitation erosivity factors of the North and South Mountains of Lanzhou in 1995, 2000, 2005, 2010, 2015, and 2018 were 110.06, 83.20, 71.09, 46.68, 56.97, and 198.61 MJ·mm/(km$^2$·h), respectively. Spatially, precipitation erosivity of the North and South Mountains decreased from southeast to northwest in 1995, 2000, 2005, and 2010, and decreased from west to east in 2015 and 2018.

(2)   The average soil erosion modulus of the North and South Mountains of Lanzhou fluctuated and decreased from 1995 to 2018. The intensity of soil erosion in 1995, 2000, 2005, 2010, 2015, and 2018 was mainly slight erosion, distributed primarily in the northwest and southeast of the North and South Mountains. Strong, extremely strong, and severe soil erosion was distributed primarily in the middle of the South Mountain, and a small amount in the middle of the North Mountain.

(3)   The soil erosion modulus of the North and South Mountains of Lanzhou increased at first and then decreased with the increase in height, increased with the increase in slope, and decreased with the increase in vegetation coverage. Among the land use types, the average soil erosion modulus of grassland and woodland was larger, and that of the water area was the lowest. The soil erosion moduli were greatest in the pedocal of the North and South Mountains, and the least in the alpine soil. The greening project effectively prevented soil erosion and initially formed an artificial ecosystem.

**Author Contributions:** Conceptualization and Methodology: H.Z.; Writing–original draft, review, and editing: J.L.; Formal analysis: H.W.; Data Curation: C.X.; Visualization: Y.Y. All authors have read and agreed to the published version of the manuscript.

**Funding:** The research was funded by the Innovation and Entrepreneurship Talent Project of Lanzhou (2019-RC-105), and the National Natural Science Foundation of China (41461011).

**Institutional Review Board Statement:** Not applicable.

**Informed Consent Statement:** The study did not involve humans.

**Data Availability Statement:** Not applicable.

**Acknowledgments:** This research received help from Chen Lei, Zhang Yuhong, An Huimin, Song Jinyue, Li Ming, and Han Wuhong from field design, sampling, and laboratory data measurement.

**Conflicts of Interest:** The funders had no role in the design of the study; in the collection, analysis, or interpretation of data; in the writing of the manuscript, or in the decision to publish the results.

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
