# Peer review of "Study on Dynamic Changes of Soil Erosion in the North and South Mountains of Lanzhou"

_water, doi:10.3390/w14152388_

Round 1

Reviewer 1 Report

This paper has analyzed the dynamic changes of soil erosion in the North and South Mountains of Lanzhou based on the RUSLE model. The workload of this study is adequate, but the analysis most are superficial. Some modification is recommended for this paper before publication. Hence, I recommend a major revision for this paper. The comments are as follows:

Major concerns:

1. Please illustrate why the North and South Mountains are chosen as the study regions, what’s the significance of the study (Introduction section)? Please add these contents to the paper (abstract section), which could attach importance to the paper.

2. The rainfall erosivity factor was calculated by an RE model based on monthly precipitation. Do you consider an RE model based on daily precipitation? Or the R factor calculation results may be could be verified by the existing studies. After verifying with existing studies, the results may be more convincing.

3. The K value is an important factor for the RUSLE model. The K factor was calculated by a formula based on the soil data measured by the laboratory. Could you illustrate how obtained the spatial distribution of the K factor in line 174? Perhaps the IDW interpolation is based on the soil samples? The K factor is a characteristic of soil, maybe it is not reasonable to interpolate by mixed soil type. Please consider more suitable methods for interpolation of the K factor.

4. Section 3.2, Too long to explain a figure, please rewrite this part, not to describe the details of the figure, just describe the main findings.

Minor comments:

1. Line 22, “Soil erosion moduli” should be modified as “Soil erosion modulus”.

2. Line 51, “1.6×108” should be modified as “1.6×108”.

3. Line 119-122, the description of elevation data may be repeated, please check.

4. Line 136, please check the equation.

5. Line 245-247, which method was adopted to obtain the value of P-value for each land use type? The relevant reference numbered [41] I have read, but the P-value for each land use type is different for cultivated land and others. Is the other land bare land? Maybe bare land is one of the main sources of soil erosion, and the bare land has no engineering measures, maybe the value is more suitable with 1.0? Please explain in the context.

6. Line 263-265, The values of annual average soil erosion should be described with the corresponding periods.

7. Line 266-271, Some sentences are repeated.

8. In figure 10, the soil erosion was higher in 2018, please explain why?

9. Line 290-291, the standard of soil erosion is “micro, moderate, strong, and extremely strong” in text, but in the figure, there are “slightly, mild, moderate, strong, extremely strong, and severe”, there are not consistent. Please be consistent or explain which standard is corresponded to other standards in the text.

10. In Table 4, the land use type (unused land) is not consistent with table 1.

11. Line 500, some blank spaces are lacking in these sentences.

Author Response

Dear Reviewer:

Re: Manuscript ID:1816578 and Title: Study on dynamic changes of soil erosion in the North and South Mountains of Lanzhou  Please see the attachment.

Thank you for your comments concerning our manuscript entitled. Those comments are valuable and very helpful. We have read through comments carefully and have made corrections. Based on the instructions provided in your letter, we uploaded the file of the revised manuscript. Revisions in the text are shown using red highlight for additions, and strikethrough font for deletions.

We would love to thank you for allowing us to resubmit a revised copy of the manuscript and we highly appreciate your time and consideration.

Sincerely.

Hua Zhang.

Q1. Please illustrate why the North and South Mountains are chosen as the study regions, what’s the significance of the study (Introduction section)? Please add these contents to the paper (abstract section), which could attach importance to the paper.

Response:We are grateful for the suggestion, the introduction and abstract section are revised.

Introduction: Lanzhou City, the capital of Gansu Province, is located in the upper basin of the  Yellow River and consists of a pearl-shaped basin formed by the alluvial deposits of the Yellow River. To the north and south of Lanzhou City are the North and South Mountains, a mountain range covered with loess formed by the terraces of the Yellow River. To the north of Lanzhou City is the Tengger Desert; to the west is the Badanjilin Desert, a region with 278.16 x 104 hm2 of severely desertified land [13], resulting in frequent sandstorms, serious surface exposure and fragile ecosystems [14-16]. The natural vegetation in the North and South Mountains of Lanzhou is mainly desert vegetation, at present, the area of the environmental greening project in the North and South Mountions of Lanzhou City has reached 413 km2, with 1.6×108 trees of various types being established, forming a complete artificial ecosystem and making the North and South Mountions an important ecological barrier [17-18]. Therefore, it is of great significance for soil and water conservation and ecological civilization construction in Lanzhou to reveal the temporal and spatial characteristics of soil erosion and analyze the dynamic changes of soil erosion in Lanzhou.

Abstract: The North and South Mountains of Lanzhou City are the ecological protection barriers  and an important part of the ecological system of Lanzhou City.This study takes the North and South Mountains as the study area, calculates the soil erosion modulus of the North and South Mountains of Lanzhou City based on the five major soil erosion factors in the RUSLE model, and analyses the spatial and temporal dynamics of soil erosion in the North and South Mountains of Lanzhou City and the soil erosion characteristics under different environmental factors. The results of the study show that: The intensity of soil erosion is dominated by slight erosion, which is dis-tributed in the northwestern and southeastern parts of the North and South Mountains in 1995, 2000, 2005, 2010, 2015 and 2018. Under different environmental factors, the soil erosion modulus in-creases with elevation and then decreases; the soil erosion modulus increases with a slope; the average soil erosion modulus of grassland is the largest, followed by forest land, cultivated land, unused land, construction land and the smallest for water; except for bare land, the average soil erosion modulus decreases with the increase of vegetation cover; Soil erosion modulus Soil ero-sion moduli are greatest in the pedocal of the North and South Mountains, and least in the alpine soil.

Q2. The rainfall erosivity factor was calculated by an RE model based on monthly precipitation. Do you consider an RE model based on daily precipitation? Or the R factor calculation results may be could be verified by the existing studies. After verifying with existing studies, the results may be more convincing.

Response: We are deeply thankful to your comment. The precipitation resources in the North and South Mountains of Lanzhou are poor and uneven, with an average annual precipitation of only 327mm, but the average annual potential evaporation is as high as 1468mm, and the precipitation is mainly concentrated from July to September, accounting for more than 60% of the annual precipitation. Therefore, we think that monthly precipitation can be well applied in northwest areas like Lanzhou. Unfortunately, we haven't found any relevant research results to prove it yet. If there is anything that can make the method more convincing later, we will make changes as soon as possible.

Q3. The K value is an important factor for the RUSLE model. The K factor was calculated by a formula based on the soil data measured by the laboratory. Could you illustrate how obtained the spatial distribution of the K factor in line 174? Perhaps the IDW interpolation is based on the soil samples? The K factor is a characteristic of soil, maybe it is not reasonable to interpolate by mixed soil type. Please consider more suitable methods for interpolation of the K factor.

Response: We are grateful to you by this suggestion. This part is our writing, which is not accurate enough. It is also explained in detail in the original text. According to the data of soil texture and soil organic carbon content of sampling points, the K value is calculated, and ordinary kriging interpolation is performed in Arcgis 10.4 software. Kringing interpolation method is suitable for spatial correlation of regionalized variables, and linear unbiased optimal estimation of unknown sample points is made according to known sample point data. Unbiased means that the mathematical expectation of deviation is 0, and optimal means that the sum of squares of the difference between the estimated value and the actual value is the smallest.

Q4. Section 3.2, Too long to explain a figure, please rewrite this part, not to describe the details of the figure, just describe the main findings.

Response: Thank you for your comments, we have rewritten this part of the content: From 1995 to 2018, the area of soil slight erosion was the largest, which was 735.92km2, followed by mild, moderate, strong, extremely strong, and the area of severe erosion remained the smallest. Among them, slight soil erosion mainly shifted to mild erosion; The mild erosion of soil transferred to slight erosion, the area was 215.14km2ï¼›Moderate soil erosion mainly shifted to slight and mild erosion; Strong soil erosion mainly shifted to mild and moderate; Moderate and strong transfer of extremely strong soil ero-sion; Severe soil erosion shifts to strong and extremely strong. From 1995 to 2018, the sta-bility rates of slight, mild, moderate, strong, extremely strong and severe soil erosion were 36.91%, 4.64%, 2.74%, 0.79%, 0.77% and 0.08% respectively (Figure 9).

Minor comments:

  1. Line 22, “Soil erosion moduli” should be modified as “Soil erosion modulus”.
  2. Line 51, “1.6×108” should be modified as “1.6×108”.
  3. Line 119-122, the description of elevation data may be repeated, please check.

Response:1-3 We corrected the mistakes according to your suggestion. Thank you for helping us to correct them.

  1. Line 136, please check the equation.

Response: We found that this is different from the article we submitted. The article we downloaded contains the complete formula. So we will take the opportunity of this revision to resubmit it.

  1. Line 245-247, which method was adopted to obtain the value of P-value for each land use type? The relevant reference numbered [41] I have read, but the P-value for each land use type is different for cultivated land and others. Is the other land bare land? Maybe bare land is one of the main sources of soil erosion, and the bare land has no engineering measures, maybe the value is more suitable with 1.0? Please explain in the context.

Response: this is important point that the date was mistake in writing. The other land is unused land, which includes bare land. We have unified the whole text, changing "other" to "unused land", at the same time, the P value of cultivated land is determined to be 0.35 by referring to other literatures.

  1. Line 263-265, The values of annual average soil erosion should be described with the corresponding periods.

Respone: “The annual average soil erosion amount is 330.74 × 104t(1995), 323.80 × 104t(2000), 342.09 × 104t(2005), 200.20 × 104t(2010), 314.41 × 104t(2015), and 515.14 × 104 t(2018).”

  1. Line 266-271, Some sentences are repeated.

Response: Thank you for your careful review. We deleted the duplicate parts.

  1. In figure 10, the soil erosion was higher in 2018, please explain why?

Response: The amount of soil erosion in 2018 was significantly higher than that in other years, mainly because 2018 was an unusually wet year, and the precipitation was higher than in previous years. Thank you for your comments, and we added them in the article.

  1. Line 290-291, the standard of soil erosion is “micro, moderate, strong, and extremely strong” in text, but in the figure, there are “slightly, mild, moderate, strong, extremely strong, and severe”, there are not consistent. Please be consistent or explain which standard is corresponded to other standards in the text.

Response: We are grateful for the suggestion. We have unified it in the article according to the division standard in the figure” slightly, mild, moderate, strong, extremely strong, and severe”

  1. In Table 4, the land use type (unused land) is not consistent with table 1.

Response: We deeply appreciate the reviewer’s suggestion, in table 1, We have changed "other" to "unused land".

  1. Line 500, some blank spaces are lacking in these sentences.

Response: Thank you for your precious comments and advice. We revised the discussion and deleted this part.

Thank you for your careful review. We really appreciate your efforts in reviewing our manuscript during this unprecedented and challenging time. We wish good health to you, your family, and community. Your careful review has helped to make our study clearer and more comprehensive.

Reviewer 2 Report

The MS 'Study on dynamic changes... mountains of Lanzhou' is well written and presented. Some minor corrections and clarifications are needed before the MS can be accepted for publication such as:

Line 37, amount of soil loss needs to be mentioned.

Line 58: few instead of low

Figures 2 and 3 can be provided as a supplementary File

Line 118: Delete (1)

Line 136: Equation is incomplete; please check

Line 137: Expand terms when cited for the first time

Line 178: Check the values?

Line 181: sentence incomplete and needs to be improved for better clarity

Line 192: Formulae and not formulas

Equation 8: not clear; state what is Beta

Units not as per SI. Please present the data in SI units. Mg/hectare. This change should be done in the entire manuscript,

Lines 269-272: some parts of the text are getting repeated. Please revise.

Use uniform style of presenting soil erosion intensity classes. Micro soil erosion cited in the text, but not presented in the Figure 11.

Lines 305-306Infact the entire paragraph including the following one needs a re-write to improve clarity.

You state in line 305; 'The unchanged area of slight soil erosion...' and then you go on to say in lines 317; 'Stable rates of soil micro,...'. If the stable rate of severe soil erosion is 0%, then how come unchanged severe soil erosion was 0.09 sq.km.

Some of your statements are self-contradictory. See the lines 328-330 and previous sentences. Therefore, linguistic improvements are needed.Lines 426-429: Incomplete statements and need to be revised. Construction land is not provided in the Table.

Author Response

Dear Reviewer:

Re: Manuscript ID:1816578 and Title: Study on dynamic changes of soil erosion in the North and South Mountains of Lanzhou Please see the attachment. 

Thank you for your comments concerning our manuscript entitled. Those comments are valuable and very helpful. We have read through comments carefully and have made corrections. Based on the instructions provided in your letter, we uploaded the file of the revised manuscript. Revisions in the text are shown using red highlight for additions, and strikethrough font for deletions.

We would love to thank you for allowing us to resubmit a revised copy of the manuscript and we highly appreciate your time and consideration.

Sincerely.

Hua Zhang.

Q1. Line 37, amount of soil loss needs to be mentioned.

Response: We deeply appreciate the reviewer’s suggestion. According to the reviewer’s comment, we have provided more details. And we found that there is no definite data about allowable soil loss, So change to: ”In 2018, the soil erosion area in China reached 2.73×106km2, accounting for about 28.80% of the total area in China except Taiwan Province”.

Q2. Line 58: few instead of low

Response: Thank you for your suggestion, we have changed ” low” to” few”

Q3. Figures 2 and 3 can be provided as a supplementary File

Response: We are grateful for the suggestion. As suggested by the reviewer, we have deleted this part from the original text and will use it as a supplementary document.

Q4. Line 118: Delete (1)

Response: The description of elevation data is repeated, we changed to” (1) The meteorological data is based on the monthly precipitation data set of 0.5 °× 0.5 °in China from 1995 to 2018 (V2.0), which comes from the China Meteorological data sharing Network (http://data.cma.cn/). (2) The GDEMDEM 30m spatial resolution digital elevation is derived from the geospatial data cloud (http://www.gscloud.cn/).”

Q5. Line 136: Equation is incomplete; please check

Response: Thank you for your suggestion. We found that this is different from the article we submitted. The article we downloaded contains the complete formula. So we will take the opportunity of this revision to resubmit it.

Q6. Line 137: Expand terms when cited for the first time

Response: Thank you for your suggestion. After introducing the formulas, they are explained one by one.

Q7. Line 178: Check the values?

Response: We are extremely grateful to reviewer for pointing out this problem. We have changed 0.45 to 0.045.

Q8. Line 181: sentence incomplete and needs to be improved for better clarity

Response: Thank you for your suggestion. we changed “the areas with a soil erodibility factor of 0.037-0.044 t·hm2×h/(hm2×MJ×mm) are mainly distributed in parts of North Mountain” to “the areas with a soil erodibility factor of 0.037-0.044 t·km2×h/(km2×MJ×mm) are mainly distributed in parts of North Mountain;”

Q9. Line 192: Formulae and not formulas

Response: Thank you for your suggestion. we changed “formulas” to “formula”.

Q10. Equation 8: not clear; state what is Beta

Response: We are extremely grateful to reviewer for pointing out this problem.β is the parameter that determines α, We have supplemented it in the article according to your suggestions.

Q11. Units not as per SI. Please present the data in SI units. Mg/hectare. This change should be done in the entire manuscript.

Response: We are grateful for the suggestion. We have unified it in the article:changed hm2 to km2.

Q12.Lines 269-272: some parts of the text are getting repeated. Please revise.

Response: Thank you for your comment, We deleted the duplicate parts.

Q13. Use uniform style of presenting soil erosion intensity classes. Micro soil erosion cited in the text, but not presented in the Figure 11.

Response: We are grateful for the suggestion. We have unified it in the article according to the division standard in the figure” slightly, mild, moderate, strong, extremely strong, and severe”.

Q14. Lines 305-306Infact the entire paragraph including the following one needs a re-write to improve clarity.

Q15. You state in line 305; 'The unchanged area of slight soil erosion...' and then you go on to say in lines 317; 'Stable rates of soil micro,...'. If the stable rate of severe soil erosion is 0%, then how come unchanged severe soil erosion was 0.09 sq.km.

Q16. Some of your statements are self-contradictory. See the lines 328-330 and previous sentences. Therefore, linguistic improvements are needed. Lines 426-429: Incomplete statements and need to be revised. Construction land is not provided in the Table.

 Response: Q14-Q16, thank you for your comments, we have rewritten this part of the content: From 1995 to 2018, the area of soil slight erosion was the largest, which was 735.92km2, followed by mild, moderate, strong, extremely strong, and the area of severe erosion remained the smallest. Among them, slight soil erosion mainly shifted to mild erosion; The mild erosion of soil transferred to slight erosion, the area was 215.14km2ï¼›Moderate soil erosion mainly shifted to slight and mild erosion; Strong soil erosion mainly shifted to mild and moderate; Moderate and strong transfer of extremely strong soil erosion; Severe soil erosion shifts to strong and extremely strong. From 1995 to 2018, the stability rates of slight, mild, moderate, strong, extremely strong and severe soil erosion were 36.91%, 4.64%, 2.74%, 0.79%, 0.77% and 0.08% respectively (Figure 9).

        We deeply appreciate the reviewer’s suggestion, in table 1, We have changed "other" to "unused land", to Unify with the table4.

Thank you for your careful review. We really appreciate your efforts in reviewing our manuscript during this unprecedented and challenging time. We wish good health to you, your family, and community. Your careful review has helped to make our study clearer and more comprehensive.

Reviewer 3 Report

The objectives of the study on dynamic changes of soil erosion in the North and South Mountains of Lanzhou was to reveal the spatial and temporal variation characteristics of soil erosion in the South and North Mountains of Lanzhou City and to provide scientific reference for the construction of water and soil conservation and ecological civilization. The research was realized using the RUSLE model.

The objectives have been finally achieved however the text of the study needs to be modified, just reconstructed. Most of the reviewer comments on that can be found in the text of the manuscript.

The Research Methods chapter is too extensive. Some parts including calculations of the RUSLE model components should be moved to the Results chapter.

Results chapter should include not only the final estimation of RUSLE model but also the calculations of its components. Methodology chapter should inform only about the methods of their determination.

Discussion is very poor. Most of the chapter instead of the discussion of obtained results with the results obtained by the other authors is devoted to recommendations of soil erosion prevention methods. The recommendations should be closely related with the obtained results and might be presented in much shorter form as the part of final conclusions.

Meanwhile, the Conclusions chapter is a summary of the obtained research results instead of indicating what the research shows.

In case of language the text should be written in the past tense instead of present tense.

Author Response

Dear Reviewer:

Re: Manuscript ID:1816578 and Title: Study on dynamic changes of soil erosion in the North and South Mountains of Lanzhou  Please see the attachment.

Thank you for your comments concerning our manuscript entitled. Those comments are valuable and very helpful. We have read through comments carefully and have made corrections. Based on the instructions provided in your letter, we uploaded the file of the revised manuscript. Revisions in the text are shown using red highlight for additions, and strikethrough font for deletions.

We would love to thank you for allowing us to resubmit a revised copy of the manuscript and we highly appreciate your time and consideration.

Sincerely.

Hua Zhang.

Q1. The Research Methods chapter is too extensive. Some parts including calculations of the RUSLE model components should be moved to the Results chapter.

Q2. Results chapter should include not only the final estimation of RUSLE model but also the calculations of its components. Methodology chapter should inform only about the methods of their determination.

Response: Thank you for your precious comments and advice. Those comments are all valuable and very helpful for revising and improving our paper. We have moved the parts about calculations of the RUSLE model components to the Results. As well as the suggestions on chapter titles in the documents you provided, we have also revised them according to it.

Q3. Discussion is very poor. Most of the chapter instead of the discussion of obtained results with the results obtained by the other authors is devoted to recommendations of soil erosion prevention methods. The recommendations should be closely related with the obtained results and might be presented in much shorter form as the part of final conclusions.

Response: We appreciate the reviewer’s positive evaluation of our work. We added a more detailed explanation about the abnormal soil erosion in 2018, and the role of plantation greening project in preventing and controlling soil erosion. The last part adds more detailed statistical analysis to the greening project.

Q4. The Conclusions chapter is a summary of the obtained research results instead of indicating what the research shows.

Response: We deeply appreciate the reviewer’s suggestion. According to the reviewer’s comment, The expression of the conclusion has been adjusted.

Q5. In case of language the text should be written in the past tense instead of present tense.

Response: We apologize for the language problems in the original manuscript. We revised the temporal problems according to the documents you provided.

Thank you for your careful review. We really appreciate your efforts in reviewing our manuscript during this unprecedented and challenging time. We wish good health to you, your family, and community. Your careful review has helped to make our study clearer and more comprehensive.

Round 2

Reviewer 1 Report

The manuscript has been properly revised, thus I recommended accepting it for publication in Water.

This manuscript is a resubmission of an earlier submission. The following is a list of the peer review reports and author responses from that submission.